# The Role of the Hercynian Mountains of Central Europe in Shaping Plant Migration Patterns in the Pleistocene—A Review

**DOI:** 10.3390/plants12183317

**Published:** 2023-09-20

**Authors:** Jacek Urbaniak, Paweł Kwiatkowski

**Affiliations:** 1Department of Botany and Plant Ecology, Wrocław University of Environmental and Life Sciences, 50-363 Wrocław, Poland; 2Institute of Biology, Biotechnology and Environmental Protection, University of Silesia in Katowice, 40-032 Katowice, Poland; pawel.kwiatkowski@us.edu.pl

**Keywords:** Hercynian mountains, Central Europe, Karkonosze, Sudetes, Erzgebirge, Bohemian Forest, phylogeography, biogeography, plant migrations, genetic structure

## Abstract

The climatic changes that took place in Europe during the Quaternary period influenced plant habitats as well as their species and vegetation composition. In this article, biogeographical studies on Hercynian mountain plants that include data for the Alps, Carpathians, and European lowlands are reviewed in order to discuss the phylogeographical structure and divergence of the Hercynian populations from those in other European mountain ranges, Scandinavia, and lowlands. The analyzed studies show specific phylogeographical relations between the Hercynian mountains, Alps, Scandinavia, Carpathians, and European lowlands. The results also indicate that the genetic patterns of plant populations in the Hercynian Mountains may differ significantly in terms of origin. The main migration routes of species to the Hercynian ranges began in the Alps or Carpathians. Some species, such as *Rubus chamaemorus* L., *Salix lapponum* L., and *Salix herbacea* L., are glacial relics that may have arrived and settled in the Hercynian Mountains during the Ice Age and that survived in isolated habitats. The Hercynian Mountains are composed of various smaller mountain ranges and are a crossroads of migration routes from different parts of Europe; thus, intensive hybridization has occurred between the plant populations therein, which is indicated by the presence of several divergent genetic lines.

## 1. Introduction

Climatic changes that took place in Europe in the Pleistocene (2.58 million years ago (Ma)—11.7 ka) strongly influenced the formation of flora and fauna [1,2,3]. This was due to successive periods of glaciations and warmer interglacial periods, which resulted in, among others, the repopulation of previously glaciated areas by plants and animals [4,5,6]. The effect of this was the transformation of plant habitats along with their species compositions and the animals inhabiting them. Some plant species could survive extremely unfavorable conditions in glacial refugia in the form of small populations in the mountains. Other species may have survived in refugia located in both southern and northern Europe, where they migrated with deteriorating climatic conditions [7,8,9]. In this way, successive glaciations and the alternating retreat of glaciers, as well as climate warming, caused plant migration, contributing to the repopulation of areas previously covered with ice or to settlement in new places [10,11]. Such climatic changes caused significant changes in gene flow patterns between populations of different plant species. These effects are also visible today in the form of broken ranges of species distribution. The current manifestation of this gene flow only reflects historical changes in the relationships between species and their populations to a small extent. However, the processes that take place in them are very important for understanding how these events from thousands of years ago shaped the current, often disjunct, geographical distributions of species, and what impact they had on the genetic variability of populations [12,13,14,15]. By comparing the evolutionary relationships of genetic lineages with their geographical distributions, we can better understand the factors that most intensively influenced genetic variability. Therefore, phylogeographic studies of organisms should always include both the temporal aspect (evolutionary relationships) and the spatial aspect (geographical range), taking into account a set of samples collected from distant sites, preferably located within the range of the entire species [12,15,16,17,18,19].

In Europe, glaciation particularly affected lowland areas north of the Alps. In the south of Europe, in lower latitudes, free from glaciers, refuges of species could have developed [12,15,20,21,22,23]. It is estimated that during the glacial periods, populations were relatively small, and a combination of genetic drift and emerging mutations influenced the level of genetic divergence between them. The main refuges in Europe during the Ice Age are believed to have been the Iberian Peninsula, the Apennine Peninsula, and the Balkans [9,24]. Comparing the genetic similarity of plant populations in refuges and other sites, we can form a clear picture of changes in the Pleistocene flora. This allows us to reconstruct the history of colonization and migration routes of species [20,25,26]. It also helps us to identify contact zones where populations from different refuges are situated close to each other [20,27,28,29]. Unfortunately, detailed studies on the Hercynian Mountains have not been the focus of mainstream phylogeographic research, and research on these mountains is lacking.

Despite some similarities, the general phylogeographic patterns of plants in Europe are not always similar. On the basis of chloroplast DNA studies of European species of trees and shrubs, significant differences in genetic variability between populations have been demonstrated [9]. They reflect the different histories of each species and type of distribution in north–south and east–west transects in Europe [20,25,26,30,31]. As a result of the presence of natural biogeographical barriers in the form of latitudinally arranged mountain ranges in Europe, which impede the free flow of genes, there is a discrepancy in gene transfer in the east–west direction compared with north–south–north migration. Related to this is another aspect of phylogeography, which is referred to as genetic biodiversity, closely related to the migration of species. During the Pleistocene, colonization by plants could have occurred from different directions and at different times. The knowledge on this subject is becoming increasingly extensive, and the development of methods and techniques of molecular analysis over the last 30 years has allowed for a completely new understanding of the issues of phylogeography and phylogeny and the related issues of evolution, inheritance, macroevolution (systematics and paleontology), and microevolution (population genetics) [20]. The main assumptions regarding the phylogeography and biogeography of organisms based on the achievements of molecular research in the 21st century have been defined in detail by [12] for animals, but are applicable to plants [15,20,32]. At the same time, the development of the first simple molecular analysis systems based on universal primers of repetitive sequences in DNA, together with the emergence of new-generation polymerases, allowed for the routine performance of analyses offering evolutionary, phylogeographic, and phylogenetic inferences. Around the end of the 1990s, very efficient and highly reproducible methods of genetic fingerprinting appeared; these included AFLP [33], which allowed for huge progress in the amount of biogeographical and phylogeographical research that could be performed. In addition to molecular biology, in order to understand the geographical genetic variability of organisms, the implementation of modern methods of spatial analysis (GIS) allowed for significant progress in phylogeographic research, including, for example, determining the factors influencing the geographical ranges of species and the genetic diversity of their populations [12,34,35].

In addition to demographic factors affecting populations, environmental, climatic, and physiographic factors are equally important for determining geographical distribution, as the basic elements of the environment directly related to the microclimatic conditions, terrain, and heterogeneity of habitats. These factors have been considered in recent decades to be the most closely responsible for the currently observed genetic biodiversity of plant populations. They are also responsible for the presence of biogeographical barriers that prevent the spread of populations of various plant species and the colonization of new places [24,26,36,37]. Therefore, the purpose of this review is (1) to present the issues of phylogeographic research carried out in the Sudetes, Bohemian Forest, and Erzgebirge, including in the Hercynian Mountains, and to explain the importance of these mountain ranges in explaining some phenomena in the field of plant biogeography; (2) to discuss the nature of the vegetation and its distribution across geographical ranges in Europe; and (3) to characterize the degree of diversity of plant populations in the context of the post-glacial spread in Europe against the background of their general patterns of disjunct distribution in the mountains, and to outline the perspectives of further phylogeographic research in the context of the analysis of plant migration patterns.

## 2. Contemporary Biogeographical Research

Contemporary biogeographical research focuses on spatially isolated populations, which developed on specific habitats in the mountains or associated with wet ecosystems (peatlands and springs). This is related to the high level of adaptation of species and entire populations to specific microclimatic conditions and local habitats. Important features shaping the complex history of species population distribution include the complex Pleistocene history of a repeated cycle of contraction of glacial refugia during cold periods and the expansion of available habitats during warm periods [38]. All this, over tens of thousands of years, resulted in limitations in the exchange of genes and taxa between isolated populations scattered in different mountain ranges during alternating periods of glaciation and warmth. Therefore, phylogeographic research is inherently multi-faceted and covers various temporal and spatial scales. This allows for a thorough analysis of mountain intra-species and population variability, as well as the genetic diversity of populations and geographical patterns of genetic lineages [38].

In relation to mountain ranges, research on the distribution and genetic variation of populations of organisms has a long history in the Alps. This has led to, among other things, the verification of many research hypotheses regarding biogeography, glacial refugia, fluctuations of plant ranges, or recolonization and migration during the Pleistocene and post-glacial periods [26,39,40]. Ref. [38] further noted that, unlike the Alps, in other mountain ranges, the biogeographical aspects of species migrations have not been extensively studied. This includes the Carpathians, Pyrenees, Apennines, Scandinavian Mountains, and Balkan Peninsula, as well as smaller mountain ranges. This group also includes the Hercynian Mountains, old mountains formed in the Paleozoic, and secondary alpine orogens that folded over time. Nevertheless, they are an important element of the European mountain system, playing a significant but relatively poorly recognized biogeographical role on the continent. Biogeographically, the Hercynian Mountains are located at the “crossroads” of European mountain ranges, ranging from Scandinavia and the Carpathians through the Alps to the Pyrenees, Apennines, and mountains of the Balkan Peninsula (Figure 1A,B).

The individual ranges of the Sudetes, especially the Krkonoše, with 26 endemics, and Hrubý Jeseník, with 9 endemics [41], are home to unique mountain flora. Among them, there are numerous glacial and boreal relics, as well as alpine taxa with a geographical distribution not generally found in Europe, limited to the Alps, Carpathians, and Scandinavia. These include, among others, *Arabis alpina* L., *Campanula barbata* L., *Carex magellanica* Lam., *Crepis sibirica* L., *Gentiana pannonica* Scop., *Hieracium villosum* Dicks., *Isoëtes lacustris* L., *Juncus trifidus* L., *Pinus mugo* Turra., *Rhododendron ferrugineum* L., *Rubus chamaemorus* L., *Salix lapponum* L., *Saxifraga nivalis* L., *Swertia perennis* L., *Trichophorum alpinum* Pers., *Vaccium microcarpum* (Turcz. ex Rupr.) Schmalh, and *Woodsia alpina* Bolton (Gray) [42,43]. Some of the populations located in the Hercynian Mountains have been included in studies analyzing the variability of plant populations, but on a much smaller scale than in the case of the previously mentioned more extensive mountain ranges in Europe (Table 1).

Phytogeographical considerations [43,78,79] partially confirm the results of the works listed in Table 1, providing a lot of interesting information on the phylogeography of vascular plants in Europe and the role of the old Hercynian Mountains in shaping the geographical ranges of plants. Many hypotheses have not yet been sufficiently substantiated by detailed research, often due to the lack of a comprehensive collection of test material, which, for many species in Europe, should include samples collected from a very wide area, ranging from Scandinavia in the north to the Alps in the south, and from the Pyrenees in the west to the Balkan Mountains located in the eastern part of the continent. The lower mountain ranges of Europe, especially those formed during the Hercynian orogeny, should also be included. Within them, there are a number of sites of species classified as glacial relicts or alpine taxa, i.e., mountain species occurring in the mountains of the Alpine system, usually above the upper forest line. In parts of the Hercynian ranges, endemic taxa are also present in the vascular flora. This applies primarily to the Karkonosze Mountains and Hrubý Jeseník, the highest mountain massifs of the Sudetes, with highly developed layers of high mountain vegetation—among others, these include *Alchemilla corcontica* Plocek; *Campanula bohemica* Hruby; *Campanula gelida* Kovanda; *Carex derelicta* Štěpánková; *Galium sudeticum* Tausch; *Knautia pseudolongifolia* Sennen; *Minuartia corcontica* Dvořáková; *Pedicularis sudetica* subsp. *sudetica* Wild.; *Poa riphaea* (Asch. and Graebn.) Fritsch; *Sorbus sudetica* Bluf, Nees, and Schauer; and numerous representatives of the genus *Hieracium* [41,80,81].

## 3. Mountains of the Hercynian Orogeny as an Important Element of the Biogeography of Organisms

The Hercynian (Variscan) orogeny is a period of intense rock-forming movements that occurred during the Palaeozoic era, between the late Silurian and the end of the Permian. These movements resulted in the formation of some of the older mountains, which are known as the Hercynids or Variscans. However, the process of their formation did not end in the Palaeozoic era, and after the end of the Permian rea, these mountains were completely eroded. Subsequently, during the Alpine orogeny, the folded Hercynian mountains were lifted a second time, as the so-called “log mountains”, and were not very high, were strongly undulating, and had a very complex geological structure, forming a permanent trace on the surface of the Earth in Central Europe in the form of a system of mountains that is partly geographically connected and that mostly constitutes remote mountain elevations such as the Meset, the Central Massife, the Vosges, the Black Forest (Schwarzwald), the Thuringian Forest, the Harz, and the Bohemian Massife: Sudetes, Erzgebirge (Krušné hory), and Bohemian Forest (Šumava) [82,83,84,85]. The features of the geological structure of the Hercynids include lithological and structural mosaicism as well as differentiated rock falls and strata and differentiated tectonics. Tectonic faults, fractures, and sinkholes are plentiful in the Hercynidian ranges. The geological structure is abundant in various different rock formations made of all types of rocks: sedimentary, magmatic, and metamorphic [86]. All of this contributes to the very high diversity of the rock substrate, both in terms of the genesis and palaeotectonic context, mainly due to the age of the Hercynidian mountain ranges [86,87]. The Hercynids are, moreover, characterized by varied altitudinal contrasts, reaching a total of about 1500 m, which formed for tens of millions of years under external environmental conditions [88,89]. This is the reason for, in addition to the varied geological relief, the highly varied geomorphological relief, which, in the case of the Sudetes, formed during the Variscan tectonic movements in the period covering the Devonian to the early Permian; thus, similar to many other massifs in the European mid-mountains [87], the characteristic geomorphological forms of the Hercynian ranges are isolated mountain ranges and massifs that are separated by basins, foothills, and upland areas [88]. Hercynian mountain massifs are mostly tectonic rifts bounded by steep erosional sills [90,91,92]. The highest parts are, in many places, flattened and mostly forested. Remnants of local glaciations throughout the Sudetes Mountain ranges include deep glacial cirques with the numerous geomorphological forms they produced. Either plate or volcanic formations are not uncommon within the Hercynian ranges, being the remnants of rather intense volcanic eruptions in the Erzgebirge, but also locally in the Sudetes Mountains. Such a rich geological history of the Hercynids together with the varied surface topography and the different climatic factors make the Hercynian ranges habitats for diverse flora.

In the context of climate change, the disjunct (island) distribution of the Hercynian ranges is equally important [86,87,88,89]. This is a completely different type than the one observed in the Pyrenees, the Alps, or the Carpathians, forming mountain ranges in Europe with a more or less latitudinal course and a length of several hundred kilometers. The effect of this is the characteristic distribution of many mountain species, whose positions are “arranged” linearly. In the case of Central Europe, the most important formation of Hercynian origin is the Bohemian Massife, which includes smaller mountain ranges; the Sudetes, including the Karkonosze and Hrubý Jeseník as their highest ranges; and the Bohemian Forest and Erzgebirge mountains. An important common feature of these units is their location on the outskirts of the Bohemian Massife (Figure 1A,B) and the presence of large swamp ecosystems (peatlands) with numerous glacial relics in the vascular flora. All of these mountain units are located quite far from much larger and much higher mountain ranges, such as the Alps or Carpathians, but they also strongly differ from them in the shapes of their vegetation floors, which are significantly lower in relation to the Carpathians and Alps, and they show a fragmentarily developed Alpine floor (only in the Karkonosze Mountains). The highest peaks of these mountain ranges reach 1605 m (Śnieżka-Karkonosze), 1491 m (Praděd-Hrubý Jeseník), 1456 m (Grosser Amber-Bohemian Forest), and 1244 m (Klínovec-Erzgebirge) above sea level. The Hercynian Mountains are dominated by elevations rarely exceeding 1000 m above sea level. Despite this, these mountains constitute a rather specific barrier in Central Europe, located at the intersection of potential north to south and east to west migration routes of organisms (Figure 1A,B). Numerous studies indicate that it may be a potential barrier to migrating plants, including high-altitude plants, and, thus, to the flow of genes between populations from both Scandinavia and Western Siberia to the south, and from the Carpathians towards the Alps and vice versa. Between Scandinavia, the Carpathians, and the Alps, there are quite extensive lowland areas, as well as the Baltic Sea, separating the Hercynian Mountains from Scandinavia; there is also the Danube Valley separating the Bohemian Forest from the Eastern Alps, and upland parts of the landscape are located between the Sudetes and the Carpathians. In their shape and spread, the Bohemian Massife mountains partly resemble the Carpathians, which also consist of smaller scattered mountain ranges located at the crossroads of migration routes. Other habitats are also important, such as peat bogs, springs or wet and calciphilous meadows, where relic species grow, including *Rubus chamaemorus* L., *Salix lapponum* L., *Swertia perennis* L., and *Trichophorum alpinum* (L.) Pers. Today, these species grow in, among other places, post-glacial cirques, nival niches, upper sections of streams, and within peat bogs. They are often remnants of the effects of the Ice Age in these mountains. Plant macrofossils and pollen records indicate that many Arctic species were widespread in the mid-altitude tundra of Central Europe during the last glaciations [25,51,93,94,95,96,97]. The rapid climate changes that occurred in the Pleistocene and later during the Holocene Thermal Maximum resulted in the expansion of forests and the elimination of much of the Arctic flora. Its remains can currently be found in Northern Europe and/or in the mountains at higher altitudes. Some Arctic species in the Hercynian ranges have survived, especially above the upper forest line in treeless tundra-like habitats, e.g., swamps, peat bogs, and spring waters. They are included in the group of glacial relics, and include, among other things, oligothermic species of extremely humid habitats, such as *Andromeda polifolia* L., *Betula nana* L., *Carex limosa* L., *Carex magellanica* Lam., *Eriophorum vaginatum L., Pedicularis sudetica* Wild., *Rubus chamaemorus* L., *Salix lapponum* L., and *Swertia perennis* L. The historical distribution of populations of various plant species in the Hercynian Mountains is in many cases disjunct. This is the reason, as in the Carpathians or the Alps, for the existence of regional refuges in Central Europe. Among other things, they determine the separate phytogeographical characteristics of the Hercynian Mountains, and are one of the elements of the phytogeographical divisions of these ranges into lower units, as in the Carpathians or the Alps [26,38]. There is not a large group of alpine and relict species in the Hercynian Mountains, as is the case in the Alps or the Carpathians. However, there are also some specific elements that testify to the distinctiveness of local flora, such as endemic species of the Karkonosze Mountains and Hrubý Jeseník, or species commonly recognized as relict, occurring in some Hercynian ranges, as well as in Scandinavia, Baltic States, and Western Siberia [98,99,100,101]. All of this contributes to the specificity of the biogeographical relations of the Hercynian Mountains (Figure 2 and Figure 3), shaped by natural abiotic factors of the environment (land relief, geological structure, water relations, and climatic conditions) and the historical development of flora and vegetation, which consists of the migrations taking place in the Pleistocene and the evolutionary processes within species over millions of years.

## 4. Biogeographic Studies in the Flora of the Hercynian Mountains

The Hercynian Mountains were not always considered interesting and important. Often, they are only minimally considered, which means that the genetic variability of European populations is incomplete. We can see from the available data that only some publications cover a wider geographic range of Hercynian populations [38,48,49,51,57,62,67,70,72,74,75] (Table 1). Only a few publications have studied the plant populations distributed in several Hercynian ranges. This applies to *Arabidopsis halleri* (L.) O’Kane et Al-Shehbaz [48,49], *Lonicera nigra* L. [62], and *Rosa pendulina* L. [74,75]. The publications presented in Table 1 provide an initial idea of what phylogeographic studies of vascular plants look like when taking into account the Hercynian belts. Possible migration routes into the region of the Hercynian mountain ranges are shown in Figure 4. Few works contain data from the Hercynids; for example, migrations from the Pleistocene refugia of *Abies alba* Mill, or *Quercus petraea* (Matt.) Liebl. and *Q. robur* L. [9,102,103,104]. These works indicate the area of the Sudetes Mountains was most probably colonized by *Abies alba* Mill. individuals derived from the refugium located in northern Italy. Chloroplastid DNA studies indicate that there is a close relationship between *Pinus mugo* Turra populations from the Hercynian mountain ranges and from the Alps through the Bohemian Forest, which is where some of the currently identified alleles probably originated [67]. In turn, the area of the Sudetes Mountains was colonized by *Arabidopisi halleri* Novon after LGM, probably by migrants from the Bohemian Forest [48].

## 5. The Origin of the Vascular Flora of Hercynian Mountains—A Review of Species

The Hercynian Mountains, although located on the outskirts of one tectonic unit (Bohemian Massife), are smaller, and, to some extent, geographically, climatically, and edaphically isolated mountain units (Figure 1A,B). Together, they comprise a spatially distant unit in relation to the Scandinavian Mountains, the Alps, or the Carpathians. Although the vegetation of the Hercynian Mountains developed in a different way from that of the Alps or Carpathians, individual mountain ranges therein do mirror the Alps, Carpathians, or Scandinavian Mountains to some extent. The number of species of common vascular plants understood as a linking element is not large, but this proves that in the past, in the periglacial and post-glacial periods, processes of flora exchange could have occurred [105]. This is especially true for mountain ranges separated by short distances, for example, between the Eastern Alps and the Bohemian Forest, and the Carpathians and Eastern Sudetes.

Although the first observations of the formation of plant ranges covering the Hercynian Mountains were made long ago [105,106], there are still many scientific questions to be answered. First of all, the processes of forming the geographic ranges of plants in the Hercynian Mountains are not sufficiently known, nor are the routes of the migration of species from other mountain ranges, after which either the settlement of the Hercynian Mountains or gene exchange took place (Figure 1A,B). In recent years, as a result of the significant intensification of research, more has become known. However, as mentioned, this knowledge remains fragmented and partial, produced through analyses of species with a wide distribution, from the Pyrenees through to the Alps, and from the Carpathians to the Balkans and subarctic areas of Europe. Based on the genetic variability of the population compared with the geographical range of plants in Europe, the probable scenarios for the spread and migration routes of species in the Postglacial period are presented below.

Wąsowicz et al. (2016) drew attention to the high haplotype heterogeneity of *Arabidopsis halleri* (L.) O’Kane et Al-Shehbaz populations from the Bohemian Forest, Harz, Karkonosze, and the Alps, and indicated the simultaneous occurrence of single, common haplotypes in mountain regions north of the Alps. This may indicate the existence of one common gene pool in the past for these areas. Of all the regions, the Bohemian Forest has the greatest genetic diversity, which may suggest the existence of a glacial refugium in this area in the past, from which migrations could potentially have taken place. Therefore, the low genetic variability of the *A. halleri* population in the Sudetes may be the result of its post-glacial origin, and the source may be the populations originating in the Bohemian Forest. Close phylogeographic relationships among the *A. halleri* from the Hercynian and Carpathian ranges have also been found [48]. The Carpathians may, thus, have served as sources of colonization in neighboring regions, including the Hercynian Mountains [49,107].

*Cicerbita alpina* Wallr. populations in Central Europe appear to be descendants of a previous extensive and continuous range resulting from Pleistocene migratory movements, and they form a cohesive genetic group found in the isolated mountain ranges of Central Europe [54]. The Sudetes, on the other hand, occupy a somewhat transitional position, as the populations in the Western Sudetes are more closely related to those in the Eastern Alps, and those in the Eastern Sudetes are closely related to those of the Western Carpathians. It is likely that some species of the Sudetes flora, including *C. alpina* populations, came from different refugia, which is supported by indicators of a genetic diversity [55].

There is a clear genetic relationship between the populations of the Western Carpathians and the Hercynian bands. This applies to, among other things, *Doronicum austriacum* Jacq., as well as to other plant species that have been studied. These indicate a close relationship between the vascular flora of the Sudetes and Western Carpathians and a potential post-glacial recolonization within the Carpathians (*Aconitum plicatum* Koehler ex Rchb. [108]; *Arabidopsis halleri*(L.) O’Kane et Al-Shehbaz [48]; *Hieracium silesiacum* E.Krasue [61]; and. *Swertia perennis* L. [77]).

The results of research on *Gentiana pannonica* Scop. indicate a minor difference between the population from the Bohemian Forest and the population from the Eastern Alps, and a more notable difference from the population from the Karkonosze Mountains [60]. These results indicate a close relationship between the Alps and the Bohemian Forest, as well as a low genetic diversity between the two regions [60,109]. These regions were probably isolated for a relatively short time, and the differences revealed between the Karkonosze Mountains and the Eastern Alps and Bohemian Forest may indicate the long-term isolation of these populations from the Karkonosze Mountains. However, there is an overlap between some samples from the Karkonosze and Bohemian Forest, which may be a consequence of the past genetic exchange between geographically distant regions [60]. Populations of *G. pannonica* from the Alps have greater genetic variability than populations from the Bohemian Forest, which may be the result of historical processes—for example, isolation and the occurrence of a so-called bottleneck in the Holocene. Certainly, the presence of a continuous subalpine zone in the Karkonosze Mountains was helpful in preserving the genetic variability of *G. pannonica* in the period of climatic optimum in this region.

The results of *Lonicera nigra* L. migration studies indicate that, despite the minimal diversity, the spread of the population of the species in the Pleistocene took place along at least two migration routes: from the Balkans and from the Carpathians. This probably resulted in the creation of a contact zone in Central Europe, in Germany. However, the populations from the Karkonosze, Izera Mountains, and Lusatian Mountains and the populations from the Carpathians are very similar, which again confirms migration in this direction, probably originating from the Carpathians [62].

Great genetic variation has been found in *Pinus mugo* Turra populations from the Alps, Sudetes, and Carpathians, and genetic similarities are only seen within mountain ranges and at nearby locations [65]. Large genetic differences between *P. mugo* populations from three distinct centers of distribution confirm the long period of spatial isolation of this species. The large distance and lack of gene exchange, as well as differences among populations in the Sudetes, Alps, and Carpathians, suggest that *P. mugo* survived the last glaciation in various refugia without contact for quite a long time [65]. There has been little gene exchange between the Carpathians (Tatra Mountains) and Sudetes (Karkonosze) populations, as well as between the Carpathians (Tatra Mountains) and the Alps, which may be the result of pollen transfer over long distances by anticyclonic circulations in the Karkonosze Mountains [110]. Winds from the south and southwest are able to carry *P. mugo* pollen from the Alps [111,112,113]. However, this kind of influence may have been much greater during the cold periods of the Pleistocene, when the geographic range of *P. mugo* covered a larger area than it does today [111,112,113].

*Pulsatilla vernalis* Mill. populations in the Sudetes (Karkonosze) and Carpathians (Tatra Mountains) were probably also isolated from the Alps, where specific haplotypes were found, indicating that these populations did not arise from a recent dispersal, but were isolated from the Alps for a long time. However, there is a closer relationship between the Tatra Mountains and Sudetes populations than is usually assumed [37,105].

AFLP analysis based on populations from the Pyrenees, Alps, and Sudetes (Karkonosze) revealed four distinct genetic groups of *Rhododendron ferrugineum* L., including one formed by the northernmost population from the Karkonosze Mountains. This strongly indicates that the population of *R. ferrugineum* in the Karkonosze Mountains is a glacial relict, and the distinctiveness and high genetic diversity indicate that it represents a previously undetected genetic lineage of the species in Europe. The species survived adverse climatic changes in a small but stable microrefugium. This is also confirmed by the fact that the Karkonosze Mountains are an important refuge for Central European mountain flora [73].

Studies of *Ranunculus platanifolius* L. covering populations from the Sudetes also indicate a very large impact of glaciations on the vegetation [114]. Modern studies using molecular markers indicate that the modern flora of the Sudetes may have originated following migration from various refugia, with the greatest affinity being with the Western Carpathians [37,45,55,61,61,74]. There is also a similarity to the Eastern Alps [114], although the results of research on *R. platanifolius* confirm the lack of significant genetic distinctiveness in the Sudeten populations, and a high similarity to the Western Carpathians. Some Sudeten populations are characterized by relatively high values of genetic diversity indices, which means that recolonization probably occurred from refugia common to the Western Carpathians and Eastern Sudetes [72]. The Scandinavian populations are part of the range of *R. platanifolius,* and have been found to be genetically related to a group of northwestern populations that includes the Western Carpathians, Alps, Dinaric Alps, Sudetes, and Pyrenees.

All *Rosa pendulina* L. populations located in the Czech Republic, including the Sudetes and the Bohemian Forest, probably come from the Carpathians, and not from the Alps as previously thought [74,78]. In addition, similar haplotypes were found in the Hrubý Jeseník and Tatra Mountains (Carpathians), where the contact zone between the Carpathian and Alpine migration routes of *R. pendulina* in the Danube valley is probably located, which likely constituted a barrier preventing alpine populations of *R. pendulina* from migrating north [74,75].

Explaining the occurrence of high-altitude flora in the relatively low ranges of the Hercynian Mountains is quite difficult. The individual ranges are clearly geographically isolated in relation to the Alpine ranges of Europe—the distance from the Sudetes to the main refuges of the Alpine and Arctic flora is large (Alps—300 km; Carpathians—300 km; subarctic regions of Scandinavia—over 1000 km). The higher parts of the Sudetes (Karkonosze and Hrubý Jeseník), forming larger areas above the forest line, constitute “islands” comparable to the alpine and subarctic ecosystems of Europe. The reason for this phenomenon may be the specific development of vegetation during the glaciation period, especially in the Pleistocene and Holocene migrations of Arctic or Arctic–Alpine species. The numerous populations in the Sudetes can be explained by pre-existing associations with the mountain flora of the Alps, Carpathians, and Scandinavia, which resulted from the functioning of the specific Arctic–Alpine tundra [115], formed during the last glaciation (Riss and Würm periods). During this period, an intensive exchange of flora in various regions of Europe took place, and an important phenomenon occurred in the movement of some species from subarctic regions to the Hercynian Mountains of Central Europe. An example is *Saxifraga nivalis* L., which, apart from subarctic areas, is only located in the Karkonosze Mountains, or some of the glacial and boreal relics (*Betula nana* L., *Pedicularis sudetica* Willd., *Rubus chamaemorus* L., and *Salix lapponum* L.). There was also a migration of species from the Alpine mountain ranges (Pyrenees, Alps, and Carpathians) through the forestless ridges of the Bohemian Massife towards the north of the continent to Scandinavia. This concerns, among others, *Campanula barbata* L., with the main center of occurrence being in the Alps, as well as several localities in Hrubý Jeseník and Śnieżnik Massife, and one occurrence in Scandinavia. *Sesleria tatrae* (Degen) Deyl thrives on the mountain ranges of the Western Carpathians, and after a range break, has been found in one isolated position in Śnieżnik Massife. Outside the Carpathians, *Thymus carpathicus* Čelak. has only been found at Hrubý Jeseník. Since the warming of the climate around the Hercynian ranges, refugia for many taxa referred to as glacial relicts were formed. Species that were widespread in the periglacial steppes during the glacial periods became taxa with divergent distribution patterns during the interglacial periods [25,35,43,79,116,117,118,119,120]. Therefore, the vertical boundaries of the distribution range have shifted many times as a result of climatic changes accompanying successive periods of glaciations, and a number of species of the Hercynian belts probably originate from different regions of Europe.

## 6. Summary

The above review of research on the origins of the flora of the Hercynian Mountains allows us to conclude that, as shown by phylogeographic studies based on molecular techniques, it is possible to solve many issues related to the biogeography of plants in this part of Europe. However, in a few studies, sufficient conclusions could be drawn from the collection of material and the solid data it yielded. However, the results obtained so far indicate that (1) the genetic patterns of plant populations in the Hercynian Mountains may differ significantly in terms of origin; (2) the main migration routes of species to the Hercynian ranges began in the Alps or Carpathians; (3) some species, such as *Rubus chamamemorus* or *Salix lapponum,* are glacial relics that may have arrived and settled in the Hercynian Mountains during the Ice Age, and survived it in isolated habitats; (4) the Hercynian Mountains, composed of various smaller mountain ranges, are a crossroads of migration routes from different part of Europe, and, thus, intensive hybridization occurs between the plant populations therein, which is indicated by the presence of several divergent genetic lines. It might be assumed that the intensive migration that took place over a period of several hundred thousand years is not so large today. The reason would primarily be the gradual lowering of the vegetation layers and the progressive climatic changes, causing ever higher temperatures in summer and winter, as well as a decrease in the amount of precipitation, which in the long run may cause the drying of habitats and the gradual disappearance of species in the high ranges of the Hercynian Mountains, characteristic of the Alps or Carpathians. An important aspect, which, in our opinion, is insufficiently discussed elsewhere, is the role of the Hercynian Mountains as a habitat colonized by plant populations from Scandinavia or Siberia. This certainly occurred, as indicated by a few studies, but due to the great distances and difficulties related to obtaining research material, the role of these regions in the recolonization of the Hercynian Mountains remains insufficiently understood. In conclusion, several aspects important for future phylogeographic research in the Hercynian Mountains have emerged from the review presented here. First of all, studies based on a wider range of species occurrence, and not limited to just a few populations from the Hercynian Ranges, Sudetes, or Bohemian Forest, are needed, which would allow for more accurate analyses and comparisons of disjunct populations from the Hercynian Mountains with populations occurring in the Alps, Carpathians, Scandinavia, Baltic Countries, or Siberia. This would allow for a more accurate determination of the phylogeographic relationships between the main biogeographic ranges. A dozen studies comparing populations from the Hercynian ranges with populations from the Alps as Carpathians is definitely not enough; there is great potential for further evolutionary research here.

## Figures and Tables

**Figure 1 plants-12-03317-f001:**
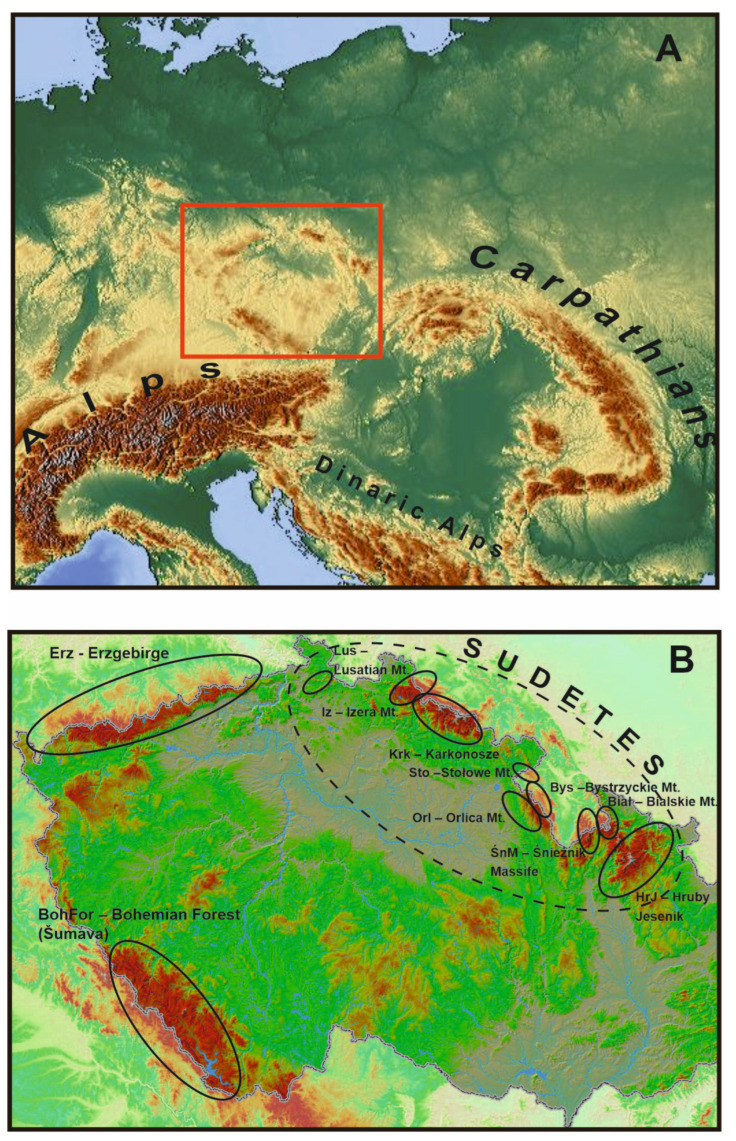
(**A**,**B**) Map of the main mountain ranges in Europe (**A**) including the area of the mountains of origin for the Hercynian orogeny, which are the subject of analysis in the article—red box (**B**). Location of mountain massifs mentioned in the text as abbreviations are as indicated.

**Figure 2 plants-12-03317-f002:**
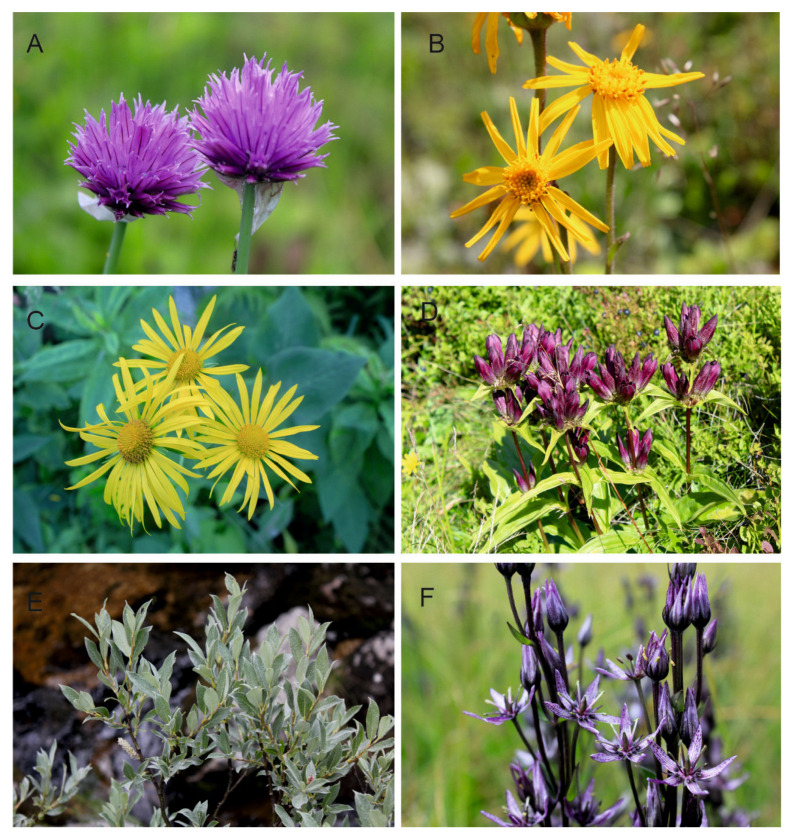
Selected mountain and boreal plants for which phylogeographic studies were carried out covering Hercynian populations: (**A**) *Allium sibiricum* L., (**B**) *Arnica montana* L., (**C**) *Doronicum austriacum* Jacq, (**D**) *Gentiana pannonica* Scop., (**E**) *Salix lapponum* L., (**F**) *Swertia perennis* L. All photos were taken by P. Kwiatkowski from Karkonosze (**A**,**E**), Erzgebirge/Krušné hory (**B**), Hruby Jesenik (**C**,**D**), and Bohemian Forest/Šumava (**F**).

**Figure 3 plants-12-03317-f003:**
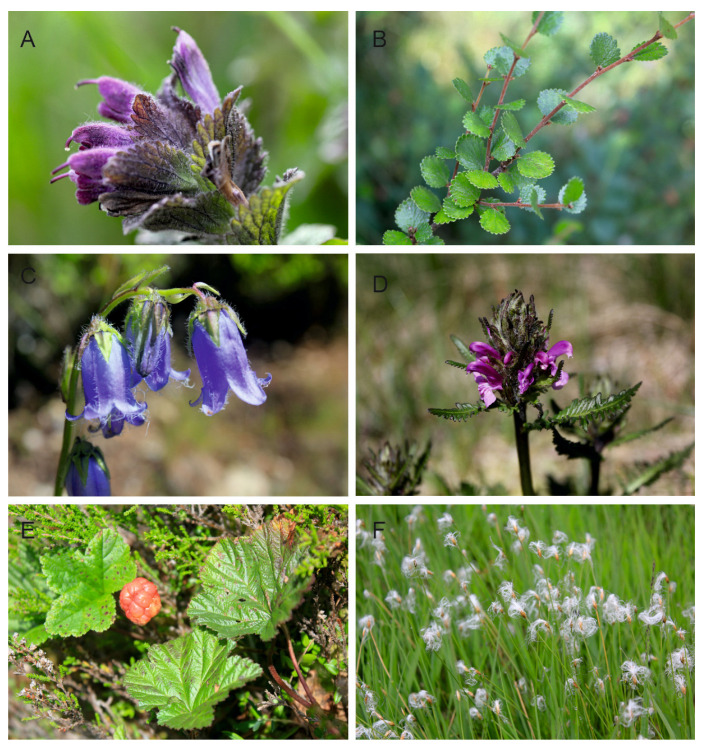
Selected endemics, relics, and alpine plants of the Hercynian massifs of Central Europe: (**A**) *Bartsia alpina* L., (**B**) *Betula nana* L., (**C**) *Campanula barbata* L., (**D**) *Pedicularis sudetica* Willd., (**E**) *Rubus chamaemorus* L., (**F**) *Trichophorum alpinum* Pers. All photos were taken by P. Kwiatkowski from Karkonosze (**A**,**D**,**E**), Bohemian Forest/Šumava (**B**,**F**), and Hruby Jesenik (**C**).

**Figure 4 plants-12-03317-f004:**
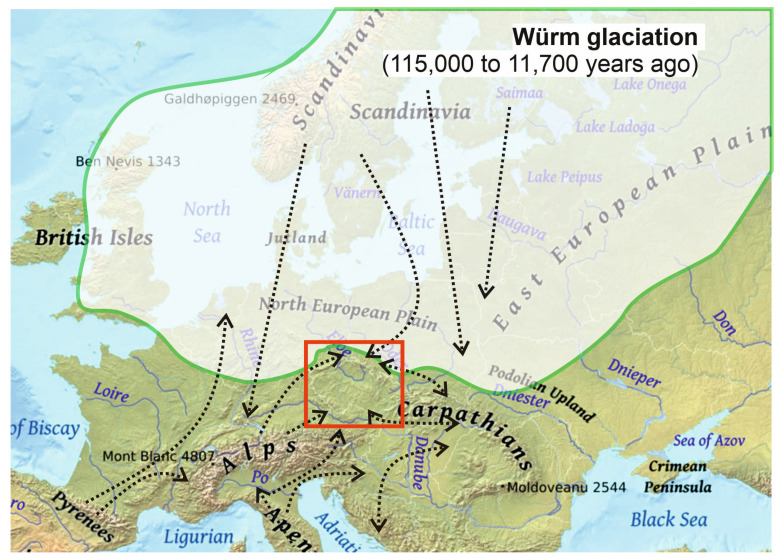
Hypothetical migration routes into the Hercynian mountains of Central Europe. The red box marks the area of the Hercynian orogeny, which is the subject of the analysis in the article.

**Table 1 plants-12-03317-t001:** Overview of phylogeographical studies that include investigations in Central Europe Mountains (Sudetes, Bohemian Forest/Šumava, and Erzgebirge/Krušné hory). Sampling abbreviations referring to the specific Sampling locality in different parts of the ranges: BohFor—Bohemian Forest/Šumava; Erz—Erzgebirge/Krušné hory; Sud—Sudetes: Bial—Góry Bialskie Mountains, Bys—Góry Bystrzyckie Mountains, HrJ—Hruby Jesenik, Iz—Izera Mountains, Krk—Karkonosze, Lus—Lusatian Mountains, ŚnM—Śnieżnik Massife, Orl—Orlica Mountains, Sto—Góry Stołowe Mountains. Sampling density estimation of the quality of population sampling as related to the species distribution (+, low; ++, medium; and +++, high). Genetic markers used in the analyses (AFLP, Amplified Fragment Length Polymorphism; cpDNA, chloroplast DNA; nDNA, nuclear DNA; mtDNA, mitochondrial DNA; ITS, Internal Transcribed Spacer of the nuclear ribosomal DNA; RAPD, Random Amplified Polymorphic DNA; PCR-RFLP, Polymerase Chain Reaction-Restriction Fragment Length Polymorphism; ISSR, Inter simple sequence repeats; cpSSR microsattelite chloroplastid DNA; NEXT, Next generation sequencing methods; Iso, Izoensymes; Cyt, flow cytometry; SNP, SNaPshot). Geographical elements: A-A—Arctic-Alpine; CB—Circum-Boreal; CE—European-temperate; ES—Euro-Siberian; IR—Irano-Turanian; M—Mediterranea.

No	Species	Family	Sampling Locality	Sampling Density	Genetic Markers Used	Geographical Scale of Study	Reference
1	*Abies alba* L.		Bys, Krk	+	cpSSR	CE	[44]
2	*Aconitum plicatum* Rchb.	Ranunculaceae	Krk	+	AFLP	CE	[45]
3	*Allium sibiricum* L.	Alliaceae	Krk	+++	ISSR	CE	[46]
4	*Anthoxanthum alpinum* A. Löve et D. Löve	Poaceae	HrJ, Krk	+	RAPD	A-A	[47]
5	*Arabidopsis halleri* (L.) O’Kane et Al-Shehbaz	Brassicaceae	BohFor, Iz, Krk, Orl	+++	SNP	ES-M-IR	[48]
6	*Arabidopsis halleri* (L.) O’Kane et Al-Shehbaz	Brassicaceae	BohFor, Iz, Krk, Bys	+++	AFLP, cpSSR	ES-M-IR	[49]
7	*Arnica montana* L.	Asteraceae	Erz	++	spSSR	CE	[50]
8	*Betula pubescens* Ehrh. subsp. *carpatica* (Willd.) Simonk	Betulaceae	BohFor, Erz, HrJ, Iz, Krk,	+++	Cut	CE	[51]
9	*Betula nana* L.	Betulaceae	Iz, Bys	+	AFLP	CB	[52]
10	*Carex bigelowii* Schwein. subsp. *rigida* W. Schultze-Motel	Cyperaceae	Krk	++	AFLP	A-A	[53]
11	*Cicerbita alpina* (L.) Wallr.	Asteraceae	Erz, Krk, ŚnM,	++	AFLP	CE	[54]
12	*Cicerbita alpina* (L.) Wallr.	Asteraceae	Bial, Krk	+++	AFLP	CE	[55]
13	*Crepis mollis* (Jacq.) Ach. s.l.	Asteraceae	Erz	++	cpSSR	CE	[56]
14	*Doronicum austriacum* Jacq.	Asteraceae	Bial, ŚnM	+++	AFLP, cpDNA, nrDNA, ITS	CE	[57]
15	*Galium anisophyllum* Vill.	Rubiaceae	ŚnM	+++	AFLP	CE	[58]
16	*Galium sudeticum* Tausch	Rubiaceae	Krk	+	AFLP	CE	[59]
17	*Galium sudeticum* Tausch	Rubiaceae	Krk	+++	AFLP	CE	[58]
18	*Gentiana pannonica* Scop.	Gentianaceae	BohFor, Krk	++	AFLP, cpDNA	CE	[60]
19	*Hieracium silesiacum* E. Krause	Asteraceae	HrJ	++	AFLP	CE	[61]
20	*Lonicera nigra* L.	Caprifoliaceae	BohFor, Iz, Lus, Krk	+++	AFLP, cpDNA	CE	[62]
21	*Meum athamanticum* Jacq.	Apiaceae	Erz	+++	AFLP	CE	[63]
22	*Meum athamanticum* Jacq.	Apiaceae	Erz	++	AFLP	CE	[64]
23	*Pinus mugo* Turra agg.	Pinaceae	Krk	+	SSR	CE	[65]
24	*Pinus mugo* Turra agg.	Pinaceae	Krk	+	cpDNA, Izo	CE	[66]
25	*Pinus mugo* Turra agg.	Pinaceae	Krk	+++	cpDNA	CE	[67]
26	*Pinus mugo* Turra agg.	Pinaceae	Sto, Bys, Krk	++	cpDNA, nDNA, mtDNA	CE	[68]
27	*Polygala chamaebuxus* L.	Polygalaceae	BohFor	++	AFLP	CE	[69]
28	*Polygonatum verticillatum* (L.) All.	Ruscaceae	Krk	+++	AFLP	A-A	[70]
29	*Primula elatior* (L.) Hill. subsp. *corcontica* (Domin) Kovanda	Primulaceae	Krk	++	cpSSR	CE	[71]
30	*Pulsatilla vernalis* L. Mill.	Ranunculaceae	Krk	+++	PCR-RFLP, cpDNA	CE	[37]
31	*Ranunculus platanifolius* L.	Ranunculaceae	Bial, ŚnM	++	AFLP	CE	[72]
32	*Rhododendron ferrugineum* L.	Ericaceae	Krk	+++	AFLP	CE	[73]
33	*Rosa pendulina* L.	Rosaceae	BohFor, HrJ, Orl, Erz	+++	cpDNA	CE	[74]
34	*Rosa pendulina* L.	Rosaceae	BohFor, HrJ, Orl, Erz	+++	AFLP	CE	[75]
35	*Rubus chamaemorus* L.	Rosaceae	Krk	++	AFLP	CB	[76]
36	*Salix lapponum* L.	Salicaceae	Krk	++	ISSR	CB	[5]
37	*Swertia perennis* L.	Gentianaceae	BohFor, HrJ, Krk	++	ISSR	CB	[77]

## Data Availability

Not applicable.

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
