# Peer review of "The Role of the Hercynian Mountains of Central Europe in Shaping Plant Migration Patterns in the Pleistocene—A Review"

_plants, 2023, doi:10.3390/plants12183317_

Round 1

Reviewer 1 Report

The authors reviewed the role of Hercynian Mountains in shaping plant migration patterns and discussed the effects linked with other mountains (e.g. Alps) in Europe. Limitation to the investigated species in this place, the authors summarized some phylogeographical patterns of plants in the Hercynian Mountains. However, such phylogeographical patterns visualized in the manuscript could be better.

Some detailed points:

Line 64: A very famous reference from Taberlet et al. (1998; Molecular Ecology) should be cited here.

Table 1: The authors should note the meanings of some abbreviations under the table, such as sampling locality and geographical scale of study.

Figure 2 & 3: Put the character A-F into each figure.

Lines 253-256: The paragraph looks a little out of place.

Line 289: From the phylogeographical studies, they could presume the migration routes of some species.

Line 273: I believe this section is the major of the manuscript. Although the authors reviewed the demographic histories of some species, I more expect that a figure illustrated plant migration patterns should be presented.

Minor editing of English language required.

Reviewer 2 Report

The ms. is interesting and covers an insufficiently investigated topic in plant biogeography. However, it should be reviewed before publication, starting with the English language. Many sentences are repeated and several passages may be shortened.

There is some problem with the organization of the text. The area focused in the ms. should be described as soon as possible and relevant geological and geomorphological information should be added. These important aspects are almost completely overlooked in the text.

The discussion could be enriched. For example, apart the mentioned examples of interesting species, is available any study about the vegetational patterns in the Hercynian Plesitocene? Could the authors provide in a schematic form any comparison with other Pleistocenic European floras? A figure with the hypothetical migration routes described in the text should be added.

English language is not satisfying and I recommend a thorough revision of the text, especially in order to avoid useless repetition and a more various and pleasant reading. Many typos can be traced throughout.

Author Response

  1. The ms. is interesting and covers an insufficiently investigated topic in plant biogeography. However, it should be reviewed before publication, starting with the English language. Many sentences are repeated and several passages may be shortened. There is some problem with the organization of the text.

Dear Reviewer,

(A). Answer:Thank you for your insightful reading and the comments contained in the attached review for our article. They are of great value to us as they allow us to improve the manuscript and enhance its quality. We have taken all the comments into account by making the appropriate corrections. These can be found below, along with explanations and an addendum to the manuscript, i.e. the new tracked and clean versions. The new version of the manuscript also includes the corrections we received in .pdf form. We reread the manuscript and tried to remove the repetitions we found in the text. We also improved the organisation of the text taking into account the comments of both reviewers. Consequently, some passages have been removed or changed. Of course, the English language has also been corrected ('clean' version) and the relevant certificate is attached.

  1. The area focused in the ms. should be described as soon as possible and relevant geological and geomorphological information should be added. These important aspects are almost completely overlooked in the text.

(A). This is an important and necessary comment. In the new version of the manuscript we have taken it into account and added basic information on geology and geomorphology. I think this will give the reader a better understanding of the specifics of the area which will of course enhance the quality of our manuscript.

  1. The discussion could be enriched. For example, apart the mentioned examples of interesting species, is available any study about the vegetational patterns in the Hercynian Plesitocene?

(A). Of course, we have taken this comment into account and supplemented the discussion with vegetational patterns in the Hercynian Plesitocene. Besides, we have partly shortened the discussion and changed it to make it more understandable for the reader.

  1. Could the authors provide in a schematic form any comparison with other Pleistocenic European floras? A figure with the hypothetical migration routes described in the text should be added.

(A). A drawing with hypothetical migrations in the Pleistocene was prepared and included in the article. It is a much-needed illustration showing possible migration routes towards the Hercynian ranges.

  1. English language is not satisfying and I recommend a thorough revision of the text, especially in order to avoid useless repetition and a more various and pleasant reading. Many typos can be traced throughout.

(A). The manuscript was corrected by the language service suggested by Plants before being sent. Of course, we did this again to eliminate linguistic errors and I hope that the text is now correct.
